# Influence of Cellulase or *Lactiplantibacillus plantarum* on the Ensiling Performance and Bacterial Community in Mixed Silage of Alfalfa and *Leymus chinensis*

**DOI:** 10.3390/microorganisms11020426

**Published:** 2023-02-08

**Authors:** Qiang Si, Zhijun Wang, Wei Liu, Mingjian Liu, Gentu Ge, Yushan Jia, Shuai Du

**Affiliations:** 1Key Laboratory of Forage Cultivation, Processing and High Efficient Utilization, Ministry of Agriculture, Hohhot 010019, China; 2Key Laboratory of Grassland Resources, Ministry of Education, Hohhot 010019, China; 3College of Grassland, Resources and Environment, Inner Mongolia Agricultural University, Hohhot 010019, China

**Keywords:** *Leymus chinensis*, *Lactiplantibacillus plantarum*, cellulase, fermentation, microbial community, mixed silages

## Abstract

The objective of this study was to evaluate the effects of *Lactiplantibacillus plantarum* or cellulase on the fermentation characteristics and bacterial community of mixed alfalfa (*Medicago sativa* L., AF) and *Leymus chinensis* (LC) silage. The harvested alfalfa and *Leymus chinensis* were cut into 1–2 cm lengths by a crop chopper and they were thoroughly mixed at a ratio of 3/2 (wet weight). The mixtures were treated with no addition (CON), *Lactiplantibacillus plantarum* (LP, 1 × 10^6^ cfu/g fresh material), cellulase (CE, 7.5 × 10^2^ U/kg fresh material) and their combination (LPCE). The forages were packed into triplicate vacuum-sealed, polyethylene bags per treatment and ensiled for 1, 3, 5, 7 and 30 d at room temperature (17–25 °C). Compared to the CON groups, all the additives increased the lactic acid content and decreased the pH and ammonia nitrogen content over the ensiling period. In comparison to the other groups, higher water-soluble carbohydrate contents were discovered in the CE-inoculated silages. Compared to the CON groups, the treatment with LPCE retained the crude protein content and reduced the acid detergent fiber content. The principal coordinate analysis based on the unweighted UniFrac distance showed that individuals in the AF, LC, CON and LPCE treatment could be significantly separated from each other. At the genus level, the bacterial community in the mixed silage involves a shift from *Cyanobacteria_unclassified* to *Lactobacillus*. *Lactobacillus* dominated in all the treatments until the end of the silage, but when added with *Lactiplantibacillus plantarum,* it was more effective in inhibiting undesirable microorganisms, such as *Enterobacter*, while reducing microbial diversity. By changing the bacterial community structure after applying *Lactiplantibacillus plantarum* and cellulase, the mixed silages quality could be further improved. During ensiling, the metabolism of the nucleotide and carbohydrate were enhanced whereas the metabolism of the amino acid, energy, cofactors and vitamins were hindered. In conclusion, the relative abundance of *Lactobacillus* in the mixed silage increased with the addition of *Lactiplantibacillus plantarum* and cellulase, which also improved the fermentation quality.

## 1. Introduction

Soil degradation is a type of complex problem facing the world that frequently takes place in arid, semi-arid and partially sub-humid regions as a result of simultaneous human activity and climatic change [1,2,3]. Horqin Sandy Land is one of the four major sandy areas in northern China [4]. For the purpose of improving the ecological environment, the growing area of alfalfa in Horqin Sandy Land is increasing since alfalfa has a high resistance to drought and salt with a rich yield and protein [5,6]. However, alfalfa can only be harvested during specific times of the year and the natural drying process leads to plenty of nutrient loss [7]. Hence, ensiling is an efficacious technique to preserve the fresh forage because it extends the storage time and improves the palatability [8]. Alfalfa has a high-buffering capacity, low dry matter (DM) and water-soluble carbohydrate (WSC) content so it often difficult to ensile [9,10,11].

In recent years, co-ensiling was recognized as an innovative method to preserve nutrition and improve the silage quality compared to single forage silage fermentation [12]. *Leymus chinensis* (*L. chinensis*) is a cool-season, perennial grass. Due to its high level of DM (260–546 g kg^−1^ FM) and protein (85–171 g kg^−1^ DM) content, it is commonly farmed on the Mongolia plateau and in western and eastern Siberian contains [13,14,15]. Consequently, co-ensiling alfalfa and *L. chinensis* may serve a number of crucial benefits, (i) alfalfa provides moisture to allow the hydrolysis of *L. chinensis*; (ii) the addition of *L. chinensis* ensures high nutrient levels and DM content in mixed silage; (iii) co-ensiling might bring about a synergistic effect on microorganisms.

In previous studies, additives were considered as green products to improve the quality of the silage fermentation [16,17]. *Lactiplantibacillus plantarum* (formerly *Lactobacillus plantarum*) was widely seen as an additive to improve the silage quality. The addition of *Lactiplantibacillus plantarum* to the high-moisture alfalfa silage improved the fermentation quality by notably raising the lactic acid content and decreasing the pH [18]. Exogenous cellulases degrade cellulose into monosaccharides or oligosaccharides, which indirectly provide fermentation substrates for lactic acid bacteria (LAB) [19]. Zhang et al. [20] indicated that adding exogenous cellulase could improve the fermentation quality of leguminous plants and reduce the complex structure of the plant cell wall. Khota et al. [21] found that cellulase enzymes can promote fiber degradation and inhibit protein hydrolysis in tropical forage feed. The combination of *Lactiplantibacillus plantarum* and cellulase has been shown to have a synergistic effect in improving the silage quality [16].

During the silage period, the microbial community showed regular changes in the different stages of silage [22]. By clarifying the changes in the microbial community structure during the silage process, it is helpful to further explore the main dominant microorganisms that affect the silage quality [23]. In recent years, silage microbial populations have been extensively studied using next-generation sequencing (NGS) [24,25]. The 16S rRNA (SSU rRNA) gene is included in the genomes of all bacteria, which is regarded as the biomarker that is used the most frequently [26]. However, current research has mainly focused on the fermentation products and microbial community changes during the silage process after the alfalfa wilts, let alone the quality and microbial community changes of the alfalfa and *L. chinensis* mixed silage. It is hypothesized that the addition of *L. plantarum* and cellulase alter the microbial community and improve the fermentation properties of a mixture of ensiled alfalfa and *L. chinensis*. Hence, the aim of this study was to evaluate the effects of *Lactiplantibacillus plantarum* and cellulase on the fermentation characteristics and microbial community dynamics of mixed (alfalfa and *L. chinensis*) silage.

## 2. Materials and Methods

### 2.1. Preparation of Silage

Alfalfa (*Medicago sativa* L., Zhongmu No. 1) and *L. chinensis* (Jisheng No. 1) were cultivated and harvested at the Inner Mongolia University for Nationalities Science and Technology Demonstration Park, Tongliao, Inner Mongolia, China (43°36′ N, 122°22′ E). The alfalfa and *Leymus chinensis* were planted on 7 June and 27 April 2018, respectively. The alfalfa was mowed three times a year, and *Leymus chinensis* was mowed once a year. Alfalfa and *Leymus chinensis* were harvested with hand sickles on 1 July 2021. Before ensiling, the harvested alfalfa and *L. chinensis* were cut into a length of 1–2 cm by a crop chopper. After that, they were thoroughly combined and blended in a 3/2 ratio (wet weight). The treatment was as follows: (i) no additive control (CON); (ii) *Lactiplantibacillus plantarum* (LP, 1 × 10^6^ colony forming units/g (cfu/g) fresh material (FM), Zhongke Jiayi Biological Engineering Co., Ltd., Shandong, China); (iii) cellulase (CE, 7.5 × 10^2^ U/kg FM, Sinopharm Chemical Reagent Co., Ltd., Shanghai, China); (iv) *Lactiplantibacillus plantarum* + cellulase (LPCE). The additives were dissolved in 20 mL of distilled water and evenly sprayed onto 600 g of the material. Then, 600 g of mixed forage was totally mixed and packed into 30 cm × 40 cm polyethylene bags (Shijiazhuang Youlang Trading Co., Ltd., Shijiazhuang, China) and vacuum sealed using a vacuum extractor (Type: DZ400/2D, Wenzhou Dafeng Machinery Co., Ltd., Wenzhou, China). A total of 60 bags (four treatments × five opening days × three replicates) were kept at room temperature (17–25 °C). The samples of 1, 3, 5, 7 and 30 d for ensiling were used to analyze the fermentation characteristics and microbial community. For the analysis of the chemical composition, the samples from Day 30 were used.

### 2.2. Chemical Component and Fermentation Characteristics Analyses

The DM content of the samples of the fresh and ensiled material were calculated after oven drying at 65 °C for 48 h [11]. The crude protein (CP) content was determined by the method of Kjeldahl [27]. According to the description of Van Soest et al. [28], the neutral detergent fiber (NDF) and acid detergent fiber (ADF) were determined by using an ANKOM A200i fiber analyzer (ANKOM Technology Corp., Fairport, NY, USA), with hemicellulose content calculated as the ADF subtracted from the NDF. The water-soluble carbohydrate (WSC) content was determined using Thomas’ method [29].

The 10 g of the silage sample was taken out, added to 90 g of distilled water and treated with a homogenizer for 2 min. The liquid extract was filtered through four layers of cheesecloth and filtered paper. The pH value was measured (LEICI pHS-3C, Shanghai, China). According to the descriptions of Cheng et al. [30], high-performance liquid chromatography was applied to determine the concentrations of lactic acid, acetic acid, propionic acid and butyric acid. The organic acid content was analyzed using an Agilent 1260 HPLC system (Agilent Technologies, Inc., Waldbronn, Germany) equipped with a refractive index detector (column: Carbomix^®^ H-NP5; Sepax Technologies, Inc., Newark, DE, USA; eluent: 2.5 mmol/L H_2_SO_4_, 0.5 mL/min; temperature: 55 °C). The ammonia nitrogen (NH_3_-N) concentration was measured using the phenol–hypochlorite method, according to the method of Broderick et al. [31].

### 2.3. DNA Extraction, PCR and Sequencing

According to the manufacturer’s instructions, the cetyltrimethylammnonium bromide (CTAB) was used to extract DNA from the sample of the fresh alfalfa, *L. chinensis*, and mixed silage. Using the universal primers 341F (5′-CCTACGGGNGGCWGCAG-3′) and 805R (5′-GACTACHVGGGTATCTAATCC-3′), bacteria 16S rDNA amplicons were sequenced. The polymerase chain reaction (PCR) amplification was carried out in LC-Bio Technology Co., Ltd., Hangzhou, China. The 2% agarose gels were used to verify the amplified PCR products. Then, the amplified PCR products were purified using the AMPure XT beads (Beckman Coulter Genomics, Danvers, MA, USA) according to the instructions of manufacturer. Lastly, the purified PCR products were quantified using Qubit^®^3.0 (Invitrogen, Carlsbad, CA, USA) and mixed evenly for each sample to create Illumina pair-end libraries in accordance with Illumina’s method for preparing genomic DNA libraries.

### 2.4. Microbial Community Analysis

The raw reads carried out quality filtering by the software of Fqtrim (v0.94) (http://ccb.jhu.edu/software/fqtrim/ (accessed on 15 September 2022)) under specific filtering conditions to gain high-quality clean tags. Vsearch software (v2.3.4) (https://github.com/torognes/vsearch (accessed on 15 September 2022)) was used for filtering the chimeric sequences. Using the Divisive Amplicon Denoising Algorithm 2 (DADA2), we acquired the feature table and feature sequence after the dereplication. In order to calculate the Alpha diversity and β-diversity, identical sequences created at random were normalized. After that, on the basis of the SILVA (version 138) (https://www.arb-silva.de/documentation/release138/ (accessed on 15 September 2022)) classifier, the relative abundance of each sample was used to normalize the feature abundance. Five indices—Chao1, Observed Species, Goods Coverage, Shannon and Simpson—were used to analyze the complexity of the species diversity for a sample. The sequence alignment was performed using Blast and the characteristic sequences of each representative sequence were annotated using the SILVA database. Based on the 16S rRNA data, the phylogenetic investigation of communities by reconstruction of unobserved states (PICRUSt) was used to predict the metabolic genes. The Kyoto Encyclopedia of Genes and Genomes (KEGG) was also used to assign genes to the metabolic pathways. The high-throughput sequencing data were analyzed on the free online platform (https://www.omicstudio.cn/ (accessed on 27 October 2022)).

### 2.5. Statistical Analyses

Data on the chemical composition, fermentation quality and microbial characteristics of the fresh and mixed silage were analyzed by the two-ways analysis of variance (ANOVA) procedure in SAS ver. 9.4 (SAS Institute, Inc., Cary, NC, USA). The statistical model is as follows:(1)$$Y_{ıjh}=\mu +\alpha_{\iota }+\beta_{j}+\alpha \beta_{ıj}+\epsilon_{ıjh}$$
where *Y_ıjh_* is an observation, *μ* is the overall mean, *α_ι_* is the effect of the additives (*ι* =CON, LP, CE, LPCE), *β_j_* is the number of ensiling days (*j* = 1, 3, 5, 7, 30), *αβ_ıj_* is the additives × the number of ensiling days interaction and *ϵ_ıjh_* is the error. The multiple comparison test by Duncan was employed. When *p* < 0.05, the effect was deemed significant. On the free online portal www.omicstudio.cn (accessed on 30 October 2022)), the high-throughput sequencing data were examined.

## 3. Results

### 3.1. Characteristics of Fresh Material

The chemical composition and microbial populations of the alfalfa and *L. chinensis* prior to ensiling are shown in Table 1. The AF was harvested at the early bloom stage. The DM, CP, NDF, ADF and WSC in the AF were 227.65 g kg^−1^ FM, 253.90 g kg^−1^ DM, 415.04 g kg^−1^ DM, 312.33 g kg^−1^ DM and 61.02 g kg^−1^ DM, respectively. The epiphytic LAB, aerobic bacteria, coliform bacteria and yeast populations of the AF were 3.87, 6.06, 4.50 and 3.49 log_10_ cfu g^−1^ FM, respectively. The DM, NDF and ADF of the LC were higher than the AF, but the CP and WSC were lower than the AF. Its content was 505.85 g kg^−1^ FM, 678.25 g kg^−1^ DM, 383.30 g kg^−1^ DM, 111.35 g kg^−1^ DM and 46.72 g kg^−1^ DM, respectively. Therefore, it is necessary to reduce the NDF content in the mixed silage of the alfalfa and *L. chinensis*. The amount of the LAB, aerobic bacteria, coliform bacteria and yeasts in the LC were 4.29, 4.29, 4.32 and 3.53 log_10_ cfu g^−1^ FM. The molds on the fresh AF and LC were below the detectable level. The mixture had a low WSC content, and the amount of the LAB was 48.70 g kg^−1^ DM and 3.74 log_10_ cfu g^−1^ FM, respectively. However, the DM content in the mixture was 335.19 g kg^−1^ FM.

### 3.2. Dynamics of Fermentation Characteristics of Mixed Silage

The fermentative characteristics, WSC and DM content of the silages are shown in the Table 2. There were two-way interactions (Treatment × Day of ensiling) for the pH, lactic acid, acetic acid, NH_3_-N and WSC that were significant (*p* < 0.01). In the whole silage process, compared to the CON silages, the LP, CE and LPCE silages had lower pH levels. The pH of the CE silage dropped fastest on Day 5 of ensiling. Throughout the ensiling period, the lactic acid concentration in the LP, CE and LPCE silages was consistently greater than that in the CON silage. The lactic acid production in the CE silage increased rapidly on Day 5 of ensiling. However, the lactic acid concentration of the LPCE silage was the highest at the end of the silage. After 3 days of ensiling, the CON silage had the greatest (*p* < 0.05) acetic acid concentration, although, throughout the three days of ensiling, the LP, CE and LPCE silages had greater (*p* < 0.05) acetic acid concentrations than the CON silage. During the ensiling procedure, the propionic acid and butyric acid were not detected in any of the treatments. The LP, CE and LPCE silages had the lower NH_3_-N concentrations compared to the CON silage. The WSC contents decreased during ensiling, while the WSC content of the other treatments decreased more rapidly on Day 5 compared to the CON silage. The CON silage had a lower (*p* < 0.05) WSC content at the end of the silage. The DM of the silage decreased with the increase in silage days, and the silage days and treatment combination had no discernible impact on the DM (*p* = 0.68).

### 3.3. Chemical Composition of Mixed Silage

Table 3 shows the chemical compositions of the 30 d silages. The CON silage had a lower (*p* < 0.05) CP content compared to the other treatments. At 30 days of silage, there was no difference in the ADF content (*p* > 0.05). The LPCE silage had a lower (*p <* 0.05) NDF content than the LP, CE and CON silages. The LPCE silage had the lowest (*p <* 0.05) hemicellulose content on Day 30 of ensiling. The DM recovery was improved (*p <* 0.05) by the addition of the LP, CE and LPCE, with the LPCE having the best DM recovery.

### 3.4. The Microbial Community of Mixed Silages during Ensiling

By focusing on the 16S rDNA variable Regions 3 and 4, high-throughput analyses were used to determine the bacterial diversity of the mixed silages in Table 4. The high-throughput amplicon sequencing generated 1,824,492 reads, and each treatment range was between 69,419 and 86,787. For all the mixed silage, the average coverage rate was more than 99%, which made it possible to analyze the microbial community. At Day 5 of ensiling, the CON silage had a higher number of Shannon index, Simpson and Chao1 values than those additive-treated silages. Compared to the CON silage, the other treatments had lower Shannon index and Simpson values at the end of ensiling, and the LP silage had the lowest Shannon index, Simpson index and Chao1 value when ensiled at 30 d.

To more clearly see if the microbial community structure in the AF, LC and silage changed, the principal coordinate analysis (PCoA) was carried out based on the weighted UniFrac distance (Figure 1). After 1, 3, 5, 7 and 30 days of silage, the bacterial diversity of the mixed silage inoculated with the additives was obviously different from that of the control through Axis 1. Furthermore, the bacterial communities of the AF, LC and 30 d of additive treatments were significantly separated. Interestingly, the LP, CE and LPCE silages were not clustered separately.

The dynamics of the bacterial communities in the fresh material and the whole silage process at the phylum level are shown (Figure 2A). The most dominant phyla in the AF were Cyanobacteria (78.36%) and Proteobacteria (18.75%). Firmicutes (46.16%) were the majority phyla of the LC, followed by Proteobacteria (43.75%) and Cyanobacteria (8.12%). The Firmicutes and Proteobacteria were the predominant phyla in the mixed silages after 30 days of ensiling (Figure 2B). Additional examination of the bacterial populations at the genus level is displayed (Figure 2C). In the present study, after 30 days of silage, *Lactobacillus* was the most prevalent genus in the silage that had been treated with *Lactiplantibacillus plantarum* (LP and LPCE; >48%) (Figure 2D). *Cyanobacteria-unclassified* was the primary epiphytic bacteria of the fresh AF. The primary bacterial genera in the LC were *Lactobacillus* (40.36%), *Cyanobacteria-unclassified* (8.12%) and *Pseudomonas* (7.64%). After 1 day of silage, *Cyanobacteria-unclassified* was the predominant genus (39.51–53.98%) among all the silage samples. Moreover, the relative abundance of in the LP silages were higher than the CON, CE and LPCE silages. With the extension of the day of ensiling, the relative abundance of *Cyanobacteria-unclassified* theatrically decreased in the CON, CE and LPCE silages after 3 days; whereas *Klebsiella* dominated the bacterial community in the CON, CE and LPCE silages. *Lactobacillus* was the most abundant genus in all the silages after 30 days of ensiling. Compared to the silage supplemented with the CE alone, the addition of the LP and LPCE reduced the abundance of *Enterobacter*. Cellulase promotes the growth of lactic acid-producing cocci (*Weissella* and *Lactococcus*). The *Weissella* in the CE (10.98%) silage was obvious. The main microbes in the CON silage were *Lactobacillus* (29.16%), *Pantoea* (15.71%), *Enterobacter* (10.30%) and *Klebsiella* (10.23%). In the LP silage, the dominant microorganism varied greatly and the most abundant genus was *Lactobacillus* (61.86%) followed by *Bacillus* (14.31%) and *Klebsiella* (7.99%). The most abundant genus in the LPCE was *Lactobacillus* (48.52%), and the relative abundance of *Cyanobacteria_unclassified* was also very high (20.94%) followed by *Bacillus* (8.64%).

Then, the differences in the bacterial communities were investigated using the LEfSe technique in four groups at different silage days (Figure 3A–D). In the CON groups (Figure 3A), the *Cyanobacteria_unclassified* and *Deltaproteobacteria_unclassified* were higher in the silage after 1 day, while *Hyphomicrobiaceae_unclassified* and *Leifsonia* were higher in the silage after 5 days, and *Lactobacillus* was high in the silage after 30 days. However, in the LP treatments (Figure 3B), among the five silage time treatments, *Lactobacillus* had the most significant difference. When the silage was at 1 day, the difference of *Weisseria* was significant. The results of the CE groups are shown in Figure 3C. *Bacillus* was higher in the CE-inoculated silages after 30 d of the ensiling period. As shown in Figure 3D, the microbial community structure in the silage inoculated with the LPCE at the genus level was stable throughout the silage period.

### 3.5. Relationships between Fermentation Parameters and Bacterial Community

Figure 4 shows the correlation between the chemical composition, fermentation parameters and bacterial community. *Lactobacillus* was negatively associated with the ADF (R^2^= −0.715, *p* < 0.01), pH (R^2^= −0.856, *p* < 0.01) and WSC (R^2^= −0.638, *p* < 0.01), but positively associated with lactic acid (R^2^= 0.682, *p* < 0.01) and acetic acid (R^2^= 0.556, *p* < 0.05). *Bacillus* showed a favorable correlation with lactic acid (R^2^= 0.650, *p* < 0.01), but a negative correlation with the pH (R^2^= −0.835, *p* < 0.01) and ADF (R^2^= −0.568, *p* < 0.05). The ADF (R^2^= 0.726, *p* < 0.01), pH (R^2^= 0.682, *p* < 0.01) and WSC (R^2^= 0.571, *p* < 0.05) were all favorably connected with *Klebsiella*, while lactic acid (R^2^= −0.565, *p* < 0.05) and acetic acid (R^2^= −0.506, *p* < 0.05) were adversely correlated. *Enterobacter* was positively correlated with the pH (R^2^= 0.606, *p* < 0.05) and WSC (R^2^= 0.521, *p* < 0.05). *Weissella* was negatively correlated with lactic acid (R^2^= −0.515, *p* < 0.05), but positively correlated with the pH (R^2^= 0.594, *p* < 0.05).

### 3.6. Bacterial Metabolic Functions Shift during Ensiling

The 16S rRNA gene prediction function spectrum describes the first, second and third pathway levels (Figure 5). On the first (Figure 5A) pathway level, the principal predicted functional genes during fermentation were distributed among the “metabolism, genetic information processing and environmental information processing” functions. The fresh samples and silage samples after 30 days at the second pathway level are shown in Figure 5B. The top 20 pathway levels in abundance were classified as human diseases (one pathway), cellular processes (two pathways), genetic information processing (four pathways), environmental information processing (two pathways) and metabolism (11 pathways). “metabolism” was obviously more prevalent relative to the other paths. Compared to other metabolic pathways, a higher relative abundance in the metabolism of carbohydrates, energy, amino acids, nucleotides, cofactors and vitamins were selected as the major metabolic pathways. At the third level of gene function prediction (Figure 5C), the abundance of amino sugar and nucleotide sugar metabolism, pentose phosphate pathway, glycolysis/gluconeogenesis and amino-acid-related enzymes genes in the LP group were higher than that in other treatments.

## 4. Discussion

### 4.1. Chemical Characteristics of Fresh Material before Ensiling

The content of the DM in the AF was much lower than 300–350 g kg^−1^ FM, the lowest DM content for producing high-quality silage [31]. After the mixing of the AF and LC in a 3:2 (wet weight) ratio, the DM of the mixed silage was 335 g kg^−1^ FM. The WSC was an important element in the silage fermentation. Silage raw materials should contain more than 5% DM and contain enough WSC to ensure successful silage fermentation [11,32]. Interestingly, the WSC content of the alfalfa was higher than that of the *L. chinensis*, which might have been caused by the growth environment of the *L. chinensis* [16]. The number of the epiphytic LAB exceeded 10^5^ cfu g^−1^of FM, and the silage was conducive to successful preservation [33,34]. As shown in Table 1, the WSC concentration in the mixture was 48.7 g kg^−1^ DM. The LAB count in the fresh AF and LC were <10^5^ cfu g^−1^ of FM. Therefore, it was vital to add *Lactiplantibacillus plantarum* or cellulase to guarantee silage success during the silage fermentation.

### 4.2. Effects of Additives on Silage Quality during Ensiling

The rate and degree of the pH decline was regarded as a key signal for the silage fermentation process reflection. In this study, all of the silages’ final pH readings fell between 4.60 and 4.90, which was regarded as sufficient for the legume silages, which typically stabilize when their pH falls to between 4.50 and 4.90 [35]. In the present study, in keeping with the findings of Desta et al. [36], it was found that the pH value decreased largely during the first 7 days of silage and then continued to decline as the silage time extended. Compared to the LP and CE silages, the LPCE silage had a lower final pH (*p* < 0.05). This might have been due to the rapid increase in lactic acid. Moreover, according to Li et al. [37], the cellulase treatment combined with the LAB was found to be more successful at lowering the pH than either the LAB or cellulase treatment alone.

At the end of ensiling, the LP silages had a lower lactic acid content than the CE silages. According to Mu et al. [17], compared to the silage added with *Lactiplantibacillus plantarum* alone, the treatment added with cellulase had a higher lactic acid content. In this study, the LPCE silage showed a higher lactic acid content and a faster pH reduction than the silage treated with *Lactiplantibacillus plantarum* or cellulase alone. The results of this test are consistent with those of Hou et al. [38], which demonstrated that the addition of *Lactiplantibacillus plantarum* and cellulase to a native grass silage significantly enhanced the fermentation quality, as seen by the minimum pH values throughout the ensiling. Cellulase can increase water-soluble carbohydrates by degrading cellulose, thus providing a fermentable substrate for *Lactiplantibacillus plantarum* to produce lactic acid. As a result, their combination had a synergistic effect in improving the quality of the silage fermentation. In the fermentation stage, the inclusion of cellulase provided improvable substrates for the intimal LAB fermentation.

Acetic acid as a hetero-fermentation product is rarely converted from lactic acid to acetic acid in the early phases of silage [39]. In this study, all the silage treatments contained small amounts of acetic acid in the early silage period. This study was consistent with previous studies reported by Schmidt et al. [40] that the acetic acid content of inoculated or uninoculated additives is basically the same in the early phase of the silage.

Mu et al. [41] reported that the high content of WSCs in the silage could delay the transition from homo-fermentation to hetero-fermentation. Thus, the concentration of acetic acid in the cellulase-added silage was significantly lower than that in the CON and LP silages. However, propionic acid was not detected in any of the treatment groups throughout the silage period. According to a study, propionic bacteria do not survive in environments with low pH [17,42] levels. Moreover, it is worth noting that Guan et al. [43] found that the reason for the undetectable propionic acid was the low fermentation capacity of the native LAB.

NH_3_-N is a significant indication of protein breakdown, which is brought on by the gradual pH decrease and the fermentation of *Clostridia* [8]. In this study, the NH_3_-N concentration in the other silages decreased compared to those of the CON. This could be due to adding *Lactiplantibacillus plantarum* and cellulase to the silage, either separately or together, that greatly reduced the NH_3_-N concentration. Moreover, *Clostridium* is considered a marker for the presence of BA and NH_3_-N in the silage, which prefers to germinate in moist environments and at high pH [44] levels. In our study, *Clostridia* and butyric acid were not detected throughout the ensiling fermentation. This might be due to the silage treatments all containing the appropriate DM content. However, the absence of butyric acid also confirmed that the rapid decrease in the pH value inhibited the fermentation of *Clostridia* [17]. Mu et al. [17] reported that adding *Lactiplantibacillus plantarum* and cellulase to the mixed silage lowered the pH, inhibiting the clostridial fermentation and resulting in the lack of BA. 

The WSC is an important substrate for the LAB fermentation, which gradually decreased during the silage process. In this study, the WSC in the LP, CE and LPCE silages was greater than the WSC in the CON silage at the ending of the silage process. According to Guo et al. [45], the silage that was supplemented with fibrolytic enzymes and *Lactiplantibacillus plantarum* preserved more WSCs so that the LAB could ferment quickly and become the dominant strain, which helped to inhibit the consumption of the WSC by the undesirable bacteria.

The silage treated with cellulase (CE and LPCE) exhibited a lower DM concentration than the silage treated with *Lactiplantibacillus plantarum*, which could be attributed to the cellulase-reducing structural carbohydrates in the silages, encouraging adequate lactic acid generation and preventing DM loss. Wang et al. [46] indicated that, compared to the silage added with *Lactiplantibacillus plantarum*, the addition of cellulase reduced the DM content in the mixed feed of the whole-plant corn and peanut vine. Moreover, the content of the CP in the LPCE silage was higher than the other treatments at 30 d of ensiling, which could be due to the fact that some silage-fermenting microbes destroy protein [47]. The silage treated with cellulase (CE and LPCE) reduced the structural carbohydrates (NDF, ADF and HC), possibly due to a combined enzymatic and acid hydrolysis effect. Mu et al. [48] discovered the content of NDF, ADF and HC were decreased, and the hydrolysis of the structural carbohydrates was accelerated by the low pH value.

### 4.3. Effects of Additives on Microbial Community Dynamics during Ensiling

In this study, Firmicutes, Proteobacteria and Cyanobacteria were found in abundance across all the treatments, and the community composition changed over time. However, it was an interesting result that we found a high abundance of Cyanobacteria in all the silages. However, according to Ogunade et al. [49], the phylum Firmicutes accounted for around 74 percent of the bacterial population in the alfalfa silage, followed by the phylum Proteobacteria and a low abundance (about one percent) of Cyanobacteria. This could be due to the fact that alfalfa grows in sandy land, resulting in different microbial community structures. This result is in line with the research of Ilyas et al. [50]. As ensiling progressed, all the silages showed that Firmicutes and Proteobacteria were the main phyla.

*Lactobacillus* is known to play a key role in the production of lactic acid to lower the pH and is often the predominant genus in a variety of high-quality silages. Likewise, *Lactobacillus* was the main genus treated with *Lactiplantibacillus plantarum* and cellulase [16]. Adding *Lactiplantibacillus plantarum* and cellulase to the silage could increase the *Lactobacillus* abundance while suppressing the undesirable microbes [46].

*Weissella* is a gram-positive, catalase-negative, heterofermentative bacterium [34]. *Weissella* is a type of LAB that is extensively utilized to improve the silage quality and plays a crucial part in the early stages of the development of silage [51]. At the end of ensiling, the relative abundance of *Weissella* in the CON, LP and LPCE groups in this study was lower than it was in the CE silage. This could be due to the independent addition of cellulase, which increases the availability of WSCs and boosts the competitiveness of *Weissella* in the silage [41].

*Enterobacter* is a facultative anaerobe that competes for fermentation substrates with LAB, producing metabolites such as NH_3_-N, succinic acid and butyric acid that impact the nutritional value and decrease the palatability [52]. In this investigation, the LP and LPCE silages had lower levels of *Enterobacter* than the CE silage. This could be explained by the fact that the addition of cellulase alone to the silage supplied a fermentation substrate, increasing the competitiveness of *Enterobacter* in the silage and leading to a larger abundance of *Enterobacter* [11].

A member of the Enterobacteriaceae family and the Proteobacteria phylum, *Klebsiella* is a facultative anaerobic bacterium that causes DM loss and produces carbon dioxide. In this study, it is intriguing to note that after 30 days of ensiling, since *Klebsiella* is a harmful bacteria that can cause cow mastitis, all the additional treatments decreased the relative abundance of *Klebsiella* compared to the CON [46]. Yuan et al. [53] was attributed to the initiation of extensive fermentation and proliferation of *Weissella*, *Lactococcus* and *Lactobacillus* to the decline of *Klebsiella* in silage.

Silage fermentation is a process leading to the change of the silage quality caused by microorganisms. In this study, *Lactobacillus* and *Bacillus* were the main microorganisms that affected the lactic acid accumulation during ensiling, which were positively correlated with the lactic acid concentration and negatively correlated with the WSC and pH. These results indicated that the WSC is consumed with the growth of microorganisms and lactic acid is generated, thus reducing the pH value [18]. *Bacillus* was negatively correlated with the ADF, which is in line with the study of Bai et al. [54]. *Bacillus* could enhance the silage fermentation, stop undesirable bacteria from growing and reduce the silage fiber content. In contrast to lactic acid, *Weissella* had a positive correlation with the pH. This might be because *Weissella* is a kind of bacteria with poor acid resistance, which grows vigorously at the initial stage of silage with higher pH values [34].

The results showed that the pH and WSC were positively correlated with *Enterobacter* and *Klebsiella*, indicating that the LAB did not dominate in the early stages of silage and that microorganisms grew and reproduced using WSCs. With the silage process, the growth and reproduction of LAB inhibited other harmful microorganisms [11]. Moreover, *Klebsiella* was positively correlated with the ADF. This might be because ADF is decomposed in the acidic environment produced by LAB. Simultaneously, the silage environment inhibited the growth of *Klebsiella*.

### 4.4. Function Shifts of Bacteria Communities in Mixed Silage

KEGG, as a new method to comprehend the function and utility of cells and organisms, evaluates the impact of microorganisms on the dynamic changes of the silage quality by predicting the functional changes of bacterial communities. Consequently, the impact of exogenous microbiota on the metabolic properties of the mixed silage were assessed using a KEGG analysis.

On the first pathway level, “metabolism” was the most common metabolic pathway (Figure 5A), which indicated that the bacteria transform into different metabolites using the fermentable substrates during the silage fermentation, resulting in a higher abundance of metabolic pathways.

The silage fermentation process is mediated by microbial activities, which degrades the substrates or transforms the metabolites through complex metabolic pathways. In this study, it was found that the silage supplemented with *Lactiplantibacillus plantarum* and cellulase could promote the metabolic capacity of WSCs. At the second pathway level, the expression of a carbohydrate metabolism pathway was related to the relative abundance of LAB in the flora. However, the abundance of the carbohydrate metabolism pathway added with cellulase was higher than that of the silage added with *Lactiplantibacillus plantarum* and cellulase, while the abundance of LAB in the silage was lower than that of the silage added with *Lactiplantibacillus plantarum* and cellulase. This might be due to the fact that the silage fermentation process that produces carbohydrates added with cellulase alone is also used by other undesirable microorganisms. Amino acids are an essential substance for plants, which are of great importance for promoting the primary metabolism and protein synthesis of plants. In this study, the amino acid metabolism was inhibited after ensiling, as reported by Wang et al. [55]. Fresh material had a lower relative abundance of “membrane transport” than the other treatments and the CON silage had the highest relative abundance of “membrane transport”. It was in line with the findings of Kilstrup et al. [56], who found that untreated silage had a higher transporter abundance. It may also be connected to the microbial communities that were attached to the two fresh materials. The LP was more abundant than other treatments in “nucleotide metabolism” and “replication and repair”. Nucleotides can be utilized to create and replicate RNA and DNA, as well as serve as the primary source of energy for biological functions, according to Wang et al. [57].

According to Bai et al. [58], the metabolism of the carbohydrates, amino acids, energy and cofactors and vitamins were the metabolic pathways connected to the silage fermentation. Therefore, some metabolic pathways that differed significantly at the second level were further analyzed at the third level. Arginine and proline metabolism and alanine, aspartate and glutamate metabolism metabolic pathways were clearly restrained after ensiling. It had to do with the quick drop in the pH levels during the initial fermentation stage, which limited the degradation of the protein in harmful microorganisms [57]. It should be noted that the metabolic path of the amino-acid-related enzymes was not significant between the fresh materials and each treatment, but the metabolic path of the amino-acid-related enzymes in each treatment was lower than that in fresh materials. It was speculated that the silage environment inhibits the metabolism of these amino acids.

The carbohydrate metabolism was mainly a glycolysis/gluconeogenesis, pentose phosphate pathway and an amino sugar and nucleotide sugar metabolism. The carbohydrate metabolism of the silage gradually increased with the storage process, and the carbohydrate metabolism pathway of the silage supplemented with *Lactiplantibacillus plantarum* was the highest. It indicated that *Lactiplantibacillus plantarum* promoted the expression of the carbohydrate metabolism pathway. Furthermore, the carbohydrate metabolism pathway of the silage added with *Lactiplantibacillus plantarum* and cellulase was weaker than that of the silage added with cellulase alone. It showed that the silage added with cellulase was utilized by undesirable microorganisms and, thus, was more involved in the expression of the carbohydrate metabolism pathway.

In this study, we found four main energy metabolism pathways, which deviated from the study of Pessione et al. [59]. The silage was lower than the fresh material in the oxidative photosynthesis, methane metabolism, photosynthesis and photosynthetic protein. The possible reason for the decrease in the abundance of photosynthesis and photosynthetic protein pathway is that the fresh materials were planted in the sand and a large amount of *Cyanobacteria* were attached to the plant surface. The low pH environment created by the LAB during silage inhibits the growth and reproduction of *Cyanobacteria*. In addition, the abundance of the oxidative photosynthesis and methane metabolism pathways in the fresh materials were greater than that in the silage of 30 days. This was because the fresh samples might have been attached with methane oxidizing bacteria. During the silage fermentation process, other bacteria used the vitamins and other growth supplements to improve the physiological conditions of the methane oxidizing bacteria [60]. This was also the reason for the metabolic pathway of metabolism of the cofactors and vitamins. In further research, we should combine the sequencing of archaea amplicons and metabolomics to clarify the mechanism of the metabolic pathway during silage.

## 5. Conclusions

The study evaluated the effects of *Lactiplantibacillus plantarum* and cellulase additives on the fermentation parameters, chemical composition and bacterial community of the mixed silage of alfalfa and *Leymus chinensis*. The results showed that the addition of *Lactiplantibacillus plantarum* and cellulase could improve the quality of the silage by rebuilding the bacterial community in the silage. The addition of *Lactiplantibacillus plantarum* and cellulase increased the abundance of *Lactobacillus*, decreased the pH value and inhibited the growth and reproduction of harmful microorganisms, such as *Enterobacter*. Compared to the addition of *Lactiplantibacillus plantarum* or cellulase alone, the addition of *Lactiplantibacillus plantarum* and cellulase retained the content of crude protein in the silage, degraded the complex structural carbohydrates and increased the concentration of lactic acid and acetic acid. In summary, adding *Lactiplantibacillus plantarum* and cellulase at the same time is a feasible strategy for alfalfa and *Leymus chinensis* mixed silage.

## Figures and Tables

**Figure 1 microorganisms-11-00426-f001:**
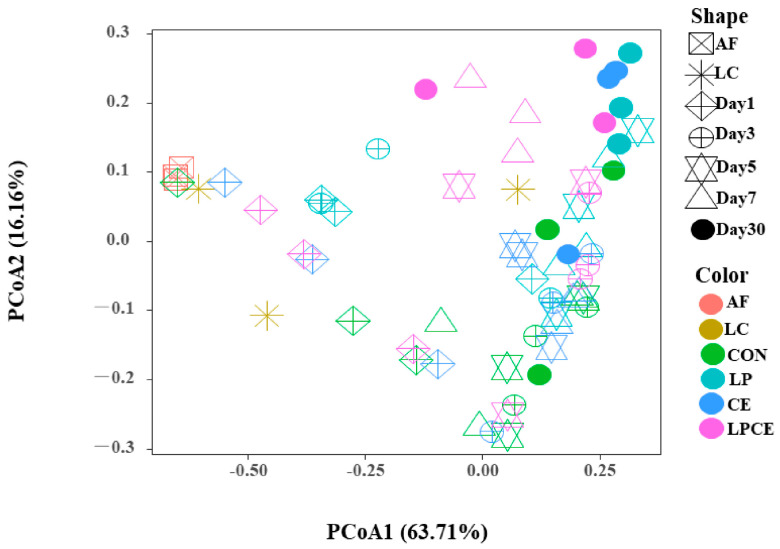
Beta diversity of the bacteria community in the mixed silage during ensiling. (n=3). The principal coordinates analysis (PCoA) of the samples conducted based on weighted UniFrac distance. AF, alfalfa; LC, *Leymus chinensis*; CON, control silage, no additive; LP, silage containing *Lactiplantibacillus plantarum*; CE, silage containing cellulase; LPCE, silage containing *Lactiplantibacillus plantarum* + cellulase.

**Figure 2 microorganisms-11-00426-f002:**
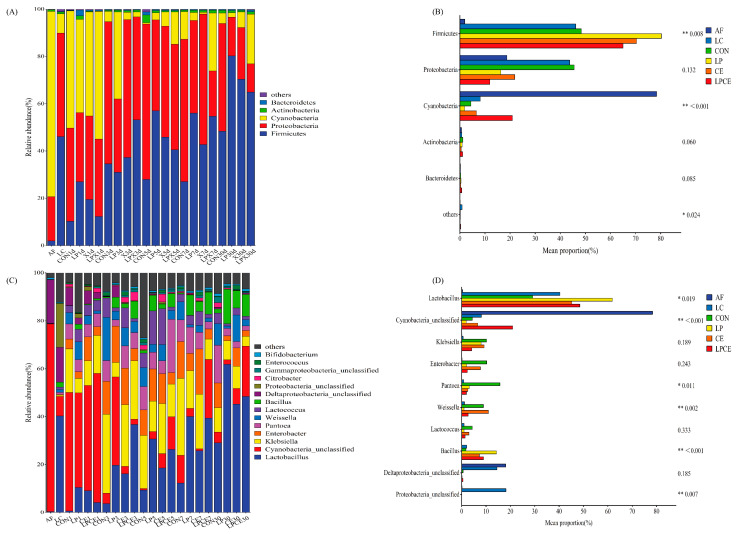
Bacterial community composition and succession in the mixed silage with or without additives. (**A**) bacterial community composition at the phylum level; (**B**) one-way analysis of variance bar plots at the phylum level among mixed groups after 30 days of silage; (**C**) bacterial community composition at the genera level; (**D**) one-way analysis of variance bar plots at the phylum level (ten most abundant genera) among the mixed groups after 30 days of silage. *, *p* < 0.05; **, *p* < 0.01; AF, fresh alfalfa; LC, *Leymus chinensis*; CON, control silage, no additive; LP, silage containing *Lactiplantibacillus plantarum*; CE, silage containing cellulase; LPCE, silage containing *Lactiplantibacillus plantarum* + cellulase.

**Figure 3 microorganisms-11-00426-f003:**
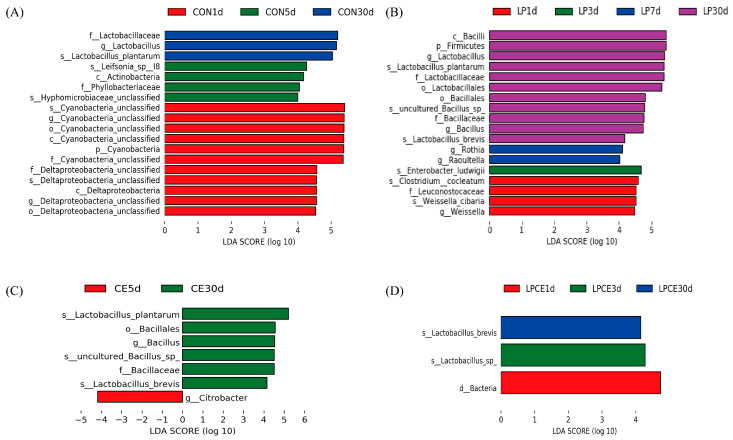
Comparison of the microbial variations using the LEfSe online tool for the mixed silage. (**A**) microbial difference of the CON group in the whole silage process; (**B**) microbial difference of the LP group in the whole silage process; (**C**) microbial difference of the CE group in the whole silage process; (**D**) microbial difference of the LPCE group in the whole silage process. CON, control silage, no additive; LP, silage containing *Lactiplantibacillus plantarum*; CE, silage containing cellulase; LPCE, silage containing *Lactiplantibacillus plantarum* + cellulase.

**Figure 4 microorganisms-11-00426-f004:**
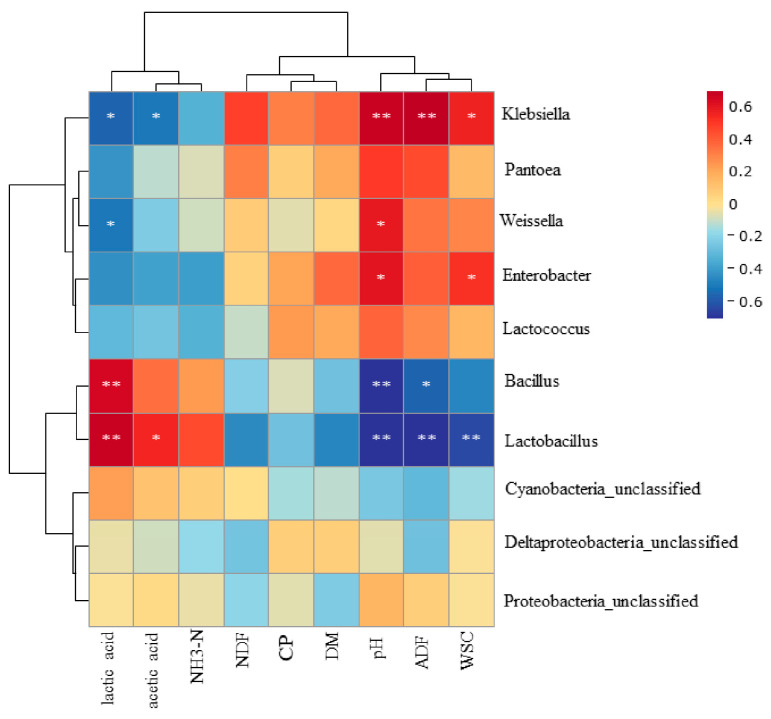
Heatmaps of Pearson’s correlations between the dominant genera, chemical compositions and fermentation quality. Red represents a positive correlation, while blue represents a negative correlation. Levels of significance are shown as follows: * *p* < 0.05; ** *p* < 0.01. NH3-N, ammonia nitrogen; NDF, neutral detergent fiber; CP, crude protein; DM, dry matter; ADF, acid detergent fiber; WSC, water-soluble carbohydrates.

**Figure 5 microorganisms-11-00426-f005:**
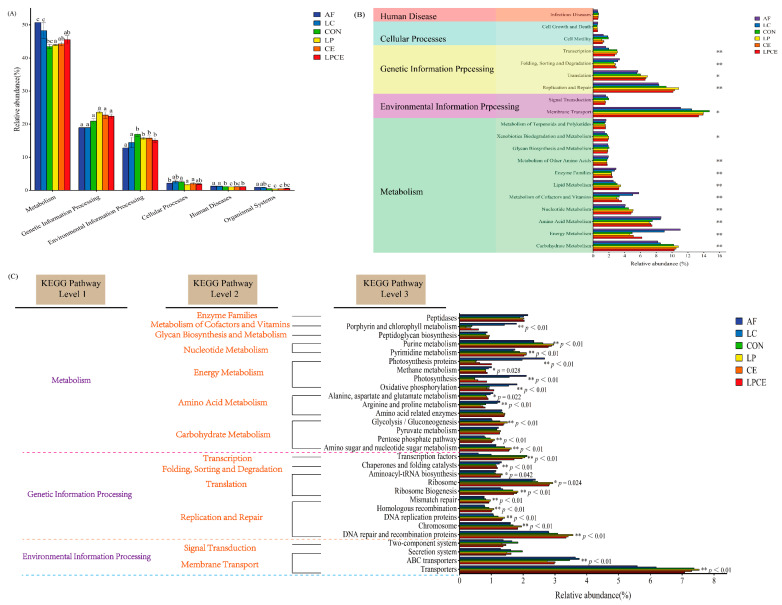
Dynamics of the bacterial functional profiles fed with different additives analyzed by PICRUSt2 (*n* = 3). (**A**) Level 1 metabolic pathways; (**B**) Level 2 Kyoto Encyclopedia of Genes and Genomes (KEGG) ortholog functional predictions of the relative abundances of the top 20 metabolic functions; (**C**) Level 3 KEGG ortholog functional predictions of the relative abundances of the top 30 metabolic functions. CON, control silage, no additive; LP, silage containing *Lactiplantibacillus plantarum*; CE, silage containing cellulase; LPCE, silage containing *Lactiplantibacillus plantarum* + cellulase. Different lowercase letters indicate significant differences among different treatments (*p* < 0.05). Levels of significant are shown as follows: * *p* < 0.05; ** *p* < 0.01.

**Table 1 microorganisms-11-00426-t001:** Chemical composition and microbial population of alfalfa (AF) and *Leymus chinensis* (LC) before ensiling.

Item	AF	LC	Mixture
Chemical composition			
Dry matter (g kg^−1^ FM)	227.65 ± 1.95	505.85 ± 3.25	335.19 ± 3.62
Crude protein (g kg^−1^ DM)	253.90 ± 2.03	111.35 ± 1.76	196.03 ± 3.51
Neutral detergent fiber (g kg^−1^ DM)	415.04 ± 1.94	678.25 ± 6.21	510.72 ± 6.34
Acid detergent fiber (g kg^−1^ DM)	312.33 ± 2.57	383.30 ± 4.57	335.71 ± 2.07
Hemicellulose (g kg^−1^ DM)	102.71 ± 1.85	294.95 ± 3.10	175.00 ± 4.27
Water-soluble carbohydrate (g kg^−1^ DM)	61.02 ± 0.69	46.72 ± 1.62	48.70 ± 0.27
Microbial population			
Lactic acid bacteria (log_10_ cfu g^−1^ FM)	3.87 ± 0.10	4.29 ± 0.06	3.74 ± 0.07
Aerobic bacteria (log_10_ cfu g^−1^ FM)	6.06 ± 0.82	4.29 ± 0.02	4.71 ± 0.05
Coliform bacteria (log cfu g^−1^ FM)	4.50 ± 0.17	4.32 ± 0.14	4.36 ± 0.08
Yeast (log_10_ cfu g^−1^ FM)	3.49 ± 0.17	3.53 ± 0.07	3.26 ± 0.14
Molds (log_10_ cfu g^−1^ FM)	ND	ND	ND

FM, fresh material; DM, dry matter; CP, crude protein; NDF, neutral detergent fiber; ADF, acid detergent fiber; WSC, water-soluble carbohydrates; LAB, lactic acid bacteria; cfu, colony forming units; ND, not detected, SEM, standard error of the mean.

**Table 2 microorganisms-11-00426-t002:** Effects of additives on mixed silage fermentative characteristics, WSC and DM contents of silages.

Item	Treatment	Ensiling Days					SEM	*p*-Value		
		1	3	5	7	30		T	D	T × D
pH	CON	5.36 ^a^	5.26 ^a^	5.21 ^a^	5.06 ^a^	4.90 ^a^	0.03	<0.01	<0.01	<0.01
	LP	5.20 ^b^	4.97 ^c^	4.87 ^b^	4.80 ^b^	4.75 ^b^				
	CE	5.15 ^bc^	5.05 ^b^	4.82 ^bc^	4.76 ^bc^	4.66 ^c^				
	LPCE	5.12 ^c^	4.95 ^c^	4.80 ^c^	4.74 ^c^	4.60 ^d^				
Lactic acid (g kg^−1^ DM)	CON	5.68 ^b^	11.06 ^b^	13.86 ^b^	21.95 ^c^	25.86 ^d^	1.19	<0.01	<0.01	<0.01
	LP	7.57 ^a^	16.50 ^a^	27.42 ^a^	26.29 ^b^	28.56 ^c^				
	CE	8.06 ^a^	11.22 ^b^	27.98 ^a^	30.06 ^a^	29.58 ^b^				
	LPCE	7.86 ^a^	16.78 ^a^	28.73 ^a^	28.83 ^a^	33.39 ^a^				
Acetic acid (g kg^−1^ DM)	CON	3.43 ^c^	4.13 ^c^	5.23 ^a^	5.61 ^a^	13.16 ^a^	0.33	<0.01	<0.01	<0.01
	LP	4.02 ^b^	4.63 ^a^	4.96 ^c^	5.36 ^b^	10.33 ^b^				
	CE	4.10 ^b^	4.58 ^a^	5.09 ^b^	5.25 ^b^	9.49 ^c^				
	LPCE	4.54 ^a^	4.42 ^b^	5.07 ^bc^	5.33 ^b^	8.54 ^d^				
Propionic acid (g kg^−1^ DM)	CON	ND	ND	ND	ND	ND				
	LP	ND	ND	ND	ND	ND				
	CE	ND	ND	ND	ND	ND				
	LPCE	ND	ND	ND	ND	ND				
Butyric acid (g kg^−1^ DM)	CON	ND	ND	ND	ND	ND				
	LP	ND	ND	ND	ND	ND				
	CE	ND	ND	ND	ND	ND				
	LPCE	ND	ND	ND	ND	ND				
NH_3_-N (g kg^–1^ TN)	CON	21.17 ^a^	23.52 ^a^	24.42 ^a^	28.02 ^a^	40.43 ^a^	0.73	<0.01	<0.01	<0.01
	LP	18.05 ^c^	18.58 ^c^	21.29 ^b^	24.42 ^c^	33.26 ^b^				
	CE	17.45 ^c^	18.44 ^c^	22.19 ^b^	25.07 ^b c^	31.20 ^c^				
	LPCE	19.15 ^b^	21.42 ^b^	23.62 ^a^	25.78 ^b^	27.71 ^d^				
WSC (g kg^−1^ DM)	CON	43.81 ^a^	34.43 ^ab^	27.23 ^a^	16.93 ^a^	6.83 ^d^	1.63	<0.01	<0.01	<0.01
	LP	38.29 ^c^	36.24 ^a^	17.83 ^b^	14.24 ^b^	7.47 ^c^				
	CE	38.47 ^c^	32.38 ^b^	15.25 ^c^	15.39 ^ab^	9.10 ^a^				
	LPCE	41.66 ^c^	36.42 ^a^	15.90 ^c^	14.20 ^b^	8.33 ^b^				
DM (g kg^−1^ FM)	CON	324.40 ^a^	321.64 ^a^	318.26 ^ab^	317.45 ^b^	309.94 ^b^	0.61	<0.01	<0.01	0.68
	LP	324.93 ^a^	322.15 ^a^	321.30 ^a^	320.80 ^a^	316.29 ^a^				
	CE	323.05 ^a^	321.09 ^a^	317.18 ^b^	314.91 ^c^	310.89 ^ab^				
	LPCE	321.74 ^a^	320.42 ^a^	317.96 ^ab^	315.55 ^bc^	311.15 ^ab^				

T, treatment; D, ensilage days; DM, dry matter; FM, fresh material; CON, control silage, no additive; LP, silage containing *Lactiplantibacillus plantarum*; CE, silage containing cellulase; LPCE, silage containing *Lactiplantibacillus plantarum* + cellulase; SEM, standard error of mean; ND, not detected; TN, total nitrogen; WSC, water-soluble carbohydrates. Different lowercase letters indicate significant differences among different silage days under the same treatment (*p* > 0.05); no or same letter indicated are not significant (*p* > 0.05).

**Table 3 microorganisms-11-00426-t003:** Effects of additives on the chemical compositions of silages after 30 days of ensiling.

Item	CP (g kg^−1^ DM)	NDF (g kg^−1^ DM)	ADF (g kg^−1^ DM)	Hemicellulose (g kg^−1^ DM)
CON	167.78 ^d^	589.61 ^a^	418.26 ^a^	171.35 ^ab^
LP	176.61 ^b^	590.55 ^a^	413.12 ^a^	177.42 ^a^
CE	171.88 ^c^	572.14 ^b^	411.62 ^a^	160.52 ^bc^
LPCE	178.49 ^a^	555.07 ^c^	406.79 ^a^	148.27 ^c^
SEM	4.40	4.52	2.29	3.75
*p*-value	<0.01	<0.01	0.43	<0.01

CP, crude protein; NDF, neutral detergent fiber; ADF, acid detergent fiber; DM, dry matter; CON, control silage, no additive; LP, silage containing *Lactiplantibacillus plantarum*; CE, silage containing cellulase; LPCE, silage containing *Lactiplantibacillus plantarum* + cellulase; SEM, standard error of means. Means in the same column (a–d) with different lowercase letters differ significantly from each other (*p* < 0.05); no or same letter indicated are not significant (*p* > 0.05).

**Table 4 microorganisms-11-00426-t004:** Effect of additives and ensiling days on bacterial alpha diversity of fresh materials and mixed silages.

Ensiling Days	Treatment	Item					
		Sequence	OTUS	Chao1	Simpson	Shannon	Coverage
FM	AF	69,419	103	109 ^c^	0.35 ^e^	1.15 ^f^	0.9998
	LC	83,855	283	288 ^ab^	0.69 ^bcd^	3.11 ^de^	0.9997
1	CON	81,978	216	222 ^bc^	0.68 ^bcd^	2.84 ^e^	0.9996
	LP	83,941	321	325 ^ab^	0.75 ^abcd^	3.91 ^bcde^	0.9996
	CE	84,979	277	285 ^ab^	0.67 ^cd^	3.20 ^cde^	0.9995
	LPCE	82,972	248	257 ^ab^	0.66 ^d^	3.10 ^de^	0.9995
3	CON	84,418	256	259 ^ab^	0.85 ^abcd^	4.01 ^acbde^	0.9996
	LP	86,787	253	259 ^ab^	0.76 ^abcd^	3.44 ^bcde^	0.9995
	CE	84,885	333	341 ^ab^	0.92 ^a^	4.98 ^ab^	0.9995
	LPCE	86,227	245	250 ^bc^	0.85 ^abcd^	4.08 ^abcde^	0.9996
5	CON	75,157	400	403 ^a^	0.92 ^a^	5.43 ^a^	0.9997
	LP	83,317	272	276 ^ab^	0.84 ^abcd^	3.84 ^bcde^	0.9996
	CE	82,243	285	291 ^ab^	0.90 ^ab^	4.59 ^abcd^	0.9996
	LPCE	85,172	249	255 ^ab^	0.81 ^abcd^	3.83 ^bcde^	0.9995
7	CON	82,408	226	230 ^bc^	0.87 ^abcd^	4.17 ^abcde^	0.9997
	LP	81,531	227	230 ^bc^	0.80 ^abcd^	3.67 ^bcde^	0.9997
	CE	83,040	292	298 ^ab^	0.89 ^abc^	4.66 ^abc^	0.9996
	LPCE	84,444	231	237 ^bc^	0.80 ^abcd^	3.60 ^bcde^	0.9996
30	CON	86,293	240	245 ^bc^	0.90 ^ab^	4.29 ^abcde^	0.9996
	LP	86,188	221	224 ^bc^	0.65 ^d^	2.88 ^e^	0.9997
	CE	63,037	255	258 ^ab^	0.83 ^abcd^	3.96 ^abcde^	0.9997
	LPCE	70,956	306	311 ^ab^	0.75 ^abcd^	3.57 ^bcde^	0.9995

FM, fresh materials; AF, alfalfa; LC, *Leymus chinensis*; CON, control silage, no additive; LP, silage containing *Lactiplantibacillus plantarum*; CE, silage containing cellulase; LPCE, silage containing *Lactiplantibacillus plantarum* + cellulase; Within a column, means without a common lowercase letter differed (*p* < 0.05).

## Data Availability

The sequencing data for the 16S rRNA gene sequence were stored at NCBI with BioProject accession number PRJNA915223.

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
