# Peer review of "Influence of Cellulase or Lactiplantibacillus plantarum on the Ensiling Performance and Bacterial Community in Mixed Silage of Alfalfa and Leymus chinensis"

_microorganisms, 2023, doi:10.3390/microorganisms11020426_

Round 1

Reviewer 1 Report

Review of “Influence of cellulase or Lactiplantibacillus plantarum on ensiling performance and bacterial community in mixed silage of alfalfa and Leymus chinensis” by Si et al.

General comments

This manuscript is of regional interest for regions that plant Leymus chinensis and has scientific merit. However, I did raise some concerns that do not allow me to recommend the manuscript for publication in its current form. Two big concerns are: (i) what is the reason for not detecting propionic acid and acetic acid, and (ii) it seems that authors ensiled the fresh material on different days.

I know it is “water under the bridge”, but why authors did not include a source of WSC (molasses, ground corn) before ensiling this material? I do not have experience with L. chinensis, but it seems not having a high content of WSC to form a good blend with alfalfa.

Although I am not a native English writer/speaker, I did realize some sentences throughout the manuscript that can be improved.

I am concerned on what authors meant by “day of ensiling”. It seems that they ensiled the fresh material on different days, which may alter chemical composition of ensiled material throughout the time points evaluated.

Abstract

I would suggest using CON as an abbreviation for the treatment with no addition of additives.

Please add somewhere in the abstract more information on experimental design and treatments. For instance, the proportion of alfalfa and LC used for ensiling, the amounts of additives added (either in IU for enzymes and cfu for Lactiplantibacillus plantarum), number of replicates per treatment, where the fresh material was ensiled (pipes? buckets?), etc.

L20: I do not think authors measured the nutritional quality of silage. The quality of silage is usually measured as a combination of chemical composition (which authors measured) and extension of degradability (often performed in vitro or in situ).

L28: p-value is not relevant in this section.

Keywords: alfalfa is already in the title no need to add as a keyword. It would be better to choose another word – for instance, enzyme or carbohydrase.

Introduction

L48: briefly explain what this region is known for – I am not from China or Mongolia, so I had to google it. “ very typical desertified grassland”

L52: I am not sure whether this citation is accurate for alfalfa. I agree that corn stalk silage has greater acceptability than fresh corn stalk, but for alfalfa authors should look for another reference.

L54-55: Delete sentence. Ensiling alfalfa is not new.

L56-57: This sentence seems to be incomplete.

L57: give more details on this plant species (grass? legume?). Maybe adding that it is a species of wild rye would be of interest to the readers.

L59: provide numbers for “high level of DM and protein content”

L65: maybe authors should state that Lactiplantibacillus plantarum was formerly named Lactobacillus plantarum.

Authors should add somewhere in the introduction section the possible advantages of using cellulase (alone) during the ensiling process of legume plants.

L75-76: the mechanism of silage…. The sentence is incomplete.

L83: review “the present study was”

Add a hypothesis.

M&M

L91-92: It is somewhat weird that plants were harvested more than 3 yr after planting.

L93: add space between 2 and cm

L94: add somewhere in this section the proportion of alfalfa and L. chinensis on a DM basis – probably the proportion of each plant was decided based on the final DM content of the mixed fresh material for ensiling.

L94-98: authors should add how treatments were applied (diluted in water/sprayed?)

L95: I am not sure whether authors had previously abbreviated fresh material as FM.

L98: Then, 600 g of mixed forage…

Please review the manuscript spaces between values and units.

L101: What authors meant by 5 ensiling days? Authors did not ensile all the material on the same day? If authors did not ensile all the fresh material on the same day there is a serious problem in M&M.

L102: Probably it is a conversion to pdf problem – Celsius degree is written in gray.

L106-107: add reference.

L113-114: Review for English.

L116: Please, add more details on silage juice extraction and reagents used before HPLC.

L117-118: I would use another abbreviation for acetic acid. AA is usually used for amino acids.

Results

L186-187: were not detected

Table 1.

Please add an abbreviation for alfalfa and Leynus chinesnsis in the title of the table

Authors should be consistent with the number of decimals in p-values.

I would rather include the chemical composition and microbial population of the mixed material instead of displaying data of AF and LC individually.

Table 2.

Acetic acid instead of acidic acid.

It is unexpected to not detect propionic and butyric acid, authors should expand the discussion on the possible causes and cite any literature that also did not detect these organic acids in silages.

Table 3.

How authors calculated DM recovery? There is no description in M&M section. The authors did not measure effluent losses or gas losses. Please, delete DM recovery data.

Some figures are blurry and difficult to read.

Discussion

L346-347: table 1 does not display the chemical composition of the mixed material.

L359: I am not sure whether LAB has been defined in the manuscript.

L369: by degrading

Authors should add somewhere what is the optimum range of pH for the cellulase used in this study.

L375-376: relative to what? Add a reference.

L381: remove p-values from the discussion section.

L398: review abbreviation (“C”).

L406: Please, expand the sentence on results found by Wang et al.

L414-471: this part of the discussion seems like a literature review. The authors should explain the results observed in this study.

Author Response

Dear Reviewer:

Thank you very much for evaluating our paper.

We thank you very much for giving us the opportunity to revise our manuscript. We appreciate editor and reviewers very much for their positive and constructive comments and suggestions on our manuscript entitled “Influence of cellulase or Lactiplantibacillus plantarum on ensiling performance and bacterial community in mixed silage of alfalfa and Leymus chinensis” (microorganisms-2143926). The comments and suggestions are not only helpful for us to revise and improve our manuscript, but also benefit our further research. We hope that our paper much better quality than before.

We would like to thank you again for taking the time to review our manuscript. If we misunderstood the comments or there are still problems with the manuscript, please let us know and we will make further explains or changes.

Best,

Qiang Si

E-Mail: siqiang_nm@126.com

Shuai Du

E-Mail: dushuai_nm@sina.com

Reviewer 2 Report

It is already well known that the addition of lactic acid bacteria and cellulase can produce good quality silage. I am not sure about the novelty. I think that it is possible to make good quality silage without adding anything (Table 2). I understand that there are significant differences, but please observe the data carefully. Is the control silage bad? We haven't observed any butyric acid or mold. Is the benefit worth the effort and cost of adding it?

Tables; If you say that superscripts are significantly different, then make them superscripts.

My specific comments are as follows.

L25; NH3-N --- What is this ?

L28; PCoA --- What is this ?

L69; LAB --- What is this ?

L88; Alfalfa --- Why don't you write the scientific name?

L95; CFU, FM --- What are these ?

L99; 30cm --- 30 cm

L100; Model number ?

L106; The dry matter (DM) --- The DM

L110; Company name and place ?

L123; CTAB --- What is this ?

L124; L. chinensis --- Italic.

L127; polymerase chain reaction --- polymerase chain reaction (PCR)

L127; conducted by "model no." (LC-Bio Technology Co., Ltd., Hangzhou, Zhejiang Province, China).

L131; Purified --- purified

L132; Company name and place ?

L136; Fqtrim --- Company name and place ?

L137; Vsearch software --- Company name and place ?

L138; DADA2 --- What is this ?

L141; SILVA --- Company name and place ?

L159; Characteristics --- Check font style.

L161-169; Isn't it unnecessary since it is written on the Table 1?

L200; Chemical --- Check font style.

L202; there was no difference in ADF content that was statistically significant (P > 0.05). --- there was no difference in ADF content (P > 0.05).

L396; lactic acid bacteria --- LAB

L400; lactic acid bacteria --- LAB

L434; lactic acid bacteria --- LAB

L441; lactic acid bacteria --- LAB

L466; lactic acid bacteria --- LAB

L468; lactic acid bacteria --- LAB

L470; lactic acid bacteria --- LAB

L473; KEGG --- Please indicate in the text (not only figure).

L486; lactic acid bacteria --- LAB

L489; lactic acid bacteria --- LAB

L493; which are be of --- Is this English all right ? Have you done native check ?

L504-539; Did you analyze them in this study? This is a redundant discussion of what the study does not reveal data on. Can you say this from the data in this study?

L530; lactic acid bacteria --- LAB

L540-548; We all know this without having to do this study.

Table 1. Chemical composition and microbial population of alfalfa and Leymus chinensis before ensiling --- Chemical composition and microbial population of alfalfa (AF) and Leymus chinensis (LC) before ensiling

What is CFU ?

Table 2. Acidic acid --- Acetic acid

Table 3.L211; means in the --- Means in the

Figure 2. I can't read the text very well.

Figure 4. What are LA, AA....ADF, and WSC ?

Figure 5. I can't read the text very well.

Author Response

(The authors gave the same response as above.)

Reviewer 3 Report

This research investigated the influence of cellulase or Lactiplantibacillus plantarum on ensiling performance and bacterial community in mixed silage of alfalfa and Leymus chinensis. This work is interesting and valuable. However, there are still several shortcomings as following.

Questions:

1. Please summarize the novelty of this research.

2. Why do you mix alfalfa and Leymus chinensis together to prepare silage?

3. About the results, are there any differences between others’ research and yours?

4. Line 169: Please check the gramma of this sentence “Molds on both fresh AF and LC below detectable levels”.

5. Please improve the quality of Figure 2 and Figure 5. The words may be too small for the audience.

Author Response

(The authors gave the same response as above.)

Reviewer 4 Report

The study called „Influence of cellulase or Lactiplantibacillus plantarum on ensiling performance and bacterial community in mixed silage of alfalfa and Leymus chinensis“ is very well written. However, some things need to be corrected.

L102: Why you did not use longer time period than 30 days? It is stated, that silage is fully stable after 60 days fermentation. Is not possible that there “something changed” after more than 30 days fermentation?

L166: How you explain lower WSC content in LC in comparison with AF? I would expect lower WSC content for alfalfa compared LC. Please explain also in discussion.

Table 2: Acidic acid is nonsense. Correct it.

Tables 2 and 3: LP is not explained – there is only L – correct it

Tables: According to description: differences are described by superscripts, but letters in tables aren´t given as superscripts.

Author Response

Dear Reviewer:

Thank you very much for evaluating our paper.

We thank you very much for giving us the opportunity to revise our manuscript. We appreciate editor and reviewers very much for their positive and constructive comments and suggestions on our manuscript entitled “Influence of cellulase or Lactiplantibacillus plantarum on ensiling performance and bacterial community in mixed silage of alfalfa and Leymus chinensis” (microorganisms-2143926). The comments and suggestions are not only helpful for us to revise and improve our manuscript, but also benefit our further research. We hope that our paper much better quality than before.

We would like to thank you again for taking the time to review our manuscript. If we misunderstood the comments or there are still problems with the manuscript, please let us know and we will make further explains or changes.

Best,

Qiang Si

E-Mail: siqiang_nm@126.com

Shuai Du

E-Mail: dushuai_nm@sina.com

Author Response File

Round 2

Reviewer 1 Report

no more comments. Authors have addressed my concerns.

Reviewer 2 Report

I have no further comments.